# Bandgap control in two-dimensional semiconductors via coherent doping of plasmonic hot electrons

Yu-Hui Chen[1,9], Ronnie R. Tamming[2,3,4,9], Kai Chen[2,3,4], Zhepeng Zhang[5], Fengjiang Liu [6,7], Yanfeng Zhang [5], Justin M. Hodgkiss[2,3,4], Richard J. Blaikie [2,3,8], Boyang Ding [2,3,8✉] & Min Qiu [6,7✉]

Bandgap control is of central importance for semiconductor technologies. The traditional means of control is to dope the lattice chemically, electrically or optically with charge carriers. Here, we demonstrate a widely tunable bandgap (renormalisation up to 550 meV at room-temperature) in two-dimensional (2D) semiconductors by coherently doping the lattice with plasmonic hot electrons. In particular, we integrate tungsten-disulfide ($WS_2$) monolayers into a self-assembled plasmonic crystal, which enables coherent coupling between semiconductor excitons and plasmon resonances. Accompanying this process, the plasmon-induced hot electrons can repeatedly fill the $WS_2$ conduction band, leading to population inversion and a significant reconstruction in band structures and exciton relaxations. Our findings provide an effective measure to engineer optical responses of 2D semiconductors, allowing flexibilities in design and optimisation of photonic and optoelectronic devices.

[1] Key Laboratory of Advanced Optoelectronic Quantum Architecture and Measurements of Ministry of Education, Beijing Key Laboratory of Nanophotonics & Ultrafine Optoelectronic Systems, School of Physics, Beijing Institute of Technology, Beijing, China. [2] Dodd-Walls Centre for Photonic and Quantum Technologies, Dunedin, New Zealand. [3] MacDiarmid Institute for Advanced Materials and Nanotechnology, Wellington, New Zealand. [4] School of Chemical and Physical Sciences, Victoria University of Wellington, Wellington, New Zealand. [5] Department of Materials Science and Engineering, College of Engineering, Center for Nanochemistry (CNC), College of Chemistry and Molecular Engineering, Academy for Advanced Interdisciplinary Studies, Peking University, Beijing, China. [6] Key Laboratory of 3D Micro/Nano Fabrication and Characterization of Zhejiang Province, School of Engineering, Westlake University, Hangzhou, Zhejiang, China. [7] Institute of Advanced Technology, Westlake Institute for Advanced Study, Hangzhou, Zhejiang, China. [8] Department of Physics, University of Otago, Dunedin, New Zealand. [9] These authors contributed equally: Yu-Hui Chen, Ronnie R. Tamming ✉email: boyang.ding@otago.ac.nz; qiu_lab@westlake.edu.cn

Two-dimensional (2D) semiconductors, such as transition metal dichalcogenides (TMDCs)[1,2], have direct bandgap at their monolayer limit, exhibiting tremendous potential in development of next-generation nanoscale devices. Like in their bulk counterparts, bandgap control plays a vital role in 2D semiconductor technologies, since it enables the creation of desirable optoelectronic properties that are required in numerous applications, ranging from lasers[3] to modulators[4], photodetectors[5], and photocatalysis[6]. The traditional means of control is to dope the lattice chemically[7], electrically[8], or optically[9] with charge carriers, the practicality of which is, however, limited by many factors, e.g., the irreversible bandgap modification, contact-type control and requirement of ultrastrong pump.

Here, we report that one can effectively modify the electronic band structures of 2D semiconductors by establishing coherent coupling between the semiconductor excitons and a plasmonic resonator[10,11]. In particular, plasmonic resonators are metallic nanostructures that support collective oscillation of electrons, known as plasmons. The excitation of plasmons can produce hot electrons, i.e., highly energetic electrons with non-equilibrium thermal distributions[12,13], which, when plasmons are coupled to 2D semiconductors, can repeatedly dope the lattice along with the coherent plasmon-exciton energy exchange. As a result, the bandgap of 2D semiconductors is significantly renormalised and the renormalisation can be easily altered via detuning plasmons from excitons.

## Results

**Sample introduction and steady-state optical properties.** The schematic of our sample in Fig. 1a demonstrates a WS$_2$ monolayer (ML) deposited onto a plasmonic crystal (PC)[14,15], which comprises of a periodic array of silver capped silica nanospheres that are coated with an ultrathin Al$_2$O$_3$ spacer with a thickness of $t = 2.5 \pm 2$ nm. This metal-insulator-semiconductor configuration constitutes PC-WS$_2$ hybrid systems, supporting plasmon lattice modes propagating on the PC-WS$_2$ interface. Here the top WS$_2$ MLs belong to the family of atomically thin TMDCs, having been extensively studied[16-20] for their unusual exciton-dominated optical responses, such as high absorption and emission efficiency. These properties make the PC-WS$_2$ systems a suitable platform to study plasmon–exciton interactions[15].

The PC geometries were chosen to excite two sets of plasmon lattice modes[10,11,14,15] (PC-01 and PC-02) that can match the frequency ($E = 2.05$ eV) of exciton A ($X_A$) and the frequency ($E = 2.42$ eV) of exciton B ($X_B$) in WS$_2$ MLs at different incident angles $\theta$. Specifically, both PC-01 and PC-02 modes show redshift dispersion at higher $\theta$ (Fig. 1b), with PC-01 being tuned in resonance with exciton A at $\theta = 22°$ and PC-02 being tuned in resonace with exciton B at $\theta = 26°$. [Please see Supplementary Figs. 1 and 2 and relevant discussions in the Supplementary Information (SI) for more details of the optical properties of bare PCs.]

As a result, within the angle range ($0 - 40°$), PC-01, PC-02, $X_A$, and $X_B$ mutually interact. To analyse the couplings between them, we have used a coupled Lorentz model that builds on four sets of Lorentz oscillators (Supplementary Eq. S1) to fit the transmission spectra of the PC-WS$_2$ sample. Figure 1c shows examples of these fittings at different incident angles. The spectral positions of the fitted resonances were then extracted and plotted as a function of angles (blue dots in Fig. 1d). The complicated dispersive behaviours of these resonances were then fitted using a ($4 \times 4$) matrix of coupled oscillators (Supplementary Eq. S2) to give the critical coupling parameters.

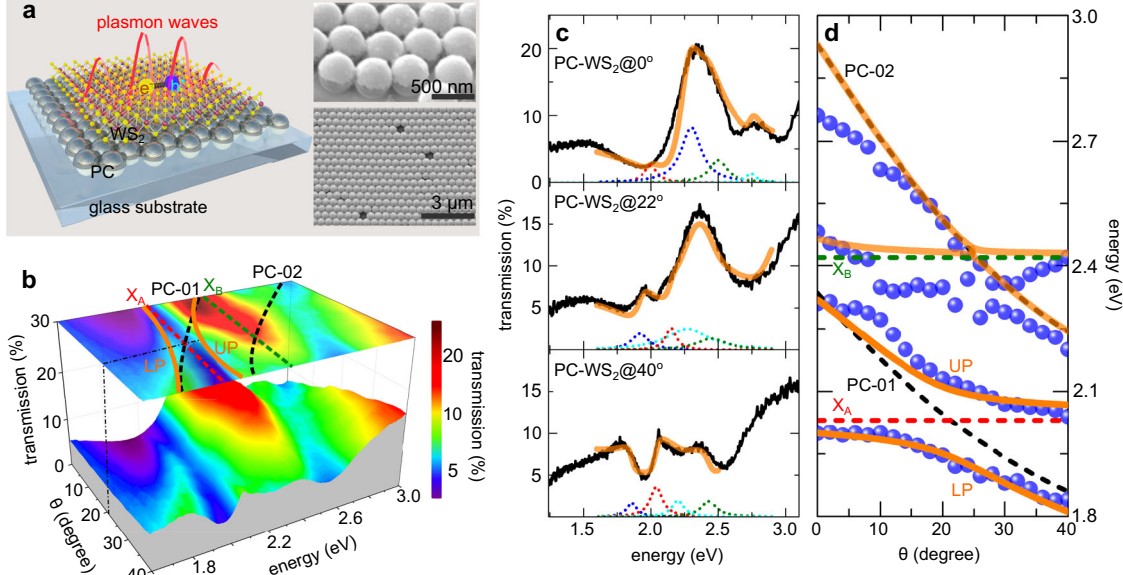

**Fig. 1 Structures of a PC-WS$_2$ sample and steady-state optical properties. a** schematic of polariton formation in a WS$_2$ ML that is supported on a self-assembled plasmonic crystal (PC), where excitons consisting of electrons (e$^-$) and holes (h$^+$) are coupled to plasmon waves. The Al$_2$O$_3$ spacer is not depicted for similicity. right insets: side and top-view scanning electron microscope images; **b** angle-resolved transmission spectra under p-polarised illumination and their projection (top x–y plane), in which the spectral positions of exciton A ($X_A$, red dashed line) and B ($X_B$, green dashed line), calculated dispersions of plasmon lattice modes (black dashed curves), and upper and lower branches of polaritons (orange curves) are indicated. The tuned angle ($\theta = 22°$) is marked with a black dash-dotted line. **c** The p-polarised transmission spectra (black curves) of a PC-WS$_2$ sample measured at $\theta = 0, 22, 40°$; thick orange curves indicate the fitted results using supplementary Eq. S1, and thin dotted curves represent the fitting components, whose amplitudes have been adjusted for visualation. **d** spectral positions of the fitted resonances (solid blue circles) in the PC-WS$_2$ sample at different illumination angles; the red (green) dashed line indicates the spectral position of exciton A (B); the black dashed curves indicate the simulated dispersions of plasmon resonances, while orange solid curves indicate the fitted dispersions using supplementary Eq. S2.

Specifically, the PC-01 mode couples with exciton A, presenting as splitting transmission maxima flanking $X_A$ (Fig. 1b, d). The spectral splitting at the tuned state ($\theta = 22°$) has a value of ~140 meV, corresponding to the plasmon-exciton coupling strength $g_{1A} \approx 87$ meV, exceeding the widely used intermediate coupling criterion $2g_{1A} > (\kappa_1 + \gamma_A)/2$[21,22], where $\kappa_1$ is the dissipation of the PC-01 mode and $\gamma_A$ is the exciton A decay rate. Such a coupling leads to the formation of plasmon–exciton polaritons, i.e., half-light half-matter quasiparticles that inherit properties from both the plasmonic and excitonic components. The frequencies of these maxima change with $\theta$, displaying dispersions that follow the upper polariton (UP) and lower polariton (LP). In contrast, the coupling between PC-02 and $X_B$ only lies in the weak coupling regime, incapable of establishing coherent energy exchange. More detailed analysis of plasmon–exciton couplings involved with other oscillators, e.g., PC-02 and $X_B$, can be found in Section 1 of SI.

**Ultrafast optical properties**. Upon photoexcitation by an optical pulse (3.1 eV and 100 fs), the transient optical responses of PC-WS$_2$ samples can be characterised using femtosecond transient absorption (TA) spectroscopy (Fig. 2a and see "Methods" section), which enables incident angle-resolved probes of the system's relaxation dynamics[23]. Figure 2b shows the tuned state ($\theta = 22°$) transient transmission spectra ($\Delta T/T$) under a pump fluence of 12 μJ cm$^{-2}$, in which the polaritonic system displays two split relaxation traces that flank the spectral position of exciton A, apparently corresponding to UP and LP. (See Supplementary Fig. 4 for discussions of transient spectral lineshapes.) This ultrashort timescale result confirms again the coherent coupling nature of the PC-WS$_2$ systems.

When the lattice plasmons are detuned from exciton A, e.g., at $\theta = 30°$, the resonances red-shift to ~1.93 eV [Supplementary Fig. 1a]. In this case, the transient spectra of PC-WS$_2$ (Fig. 2c) only show a single relaxation trace at ~2.08 eV, which, by comparing

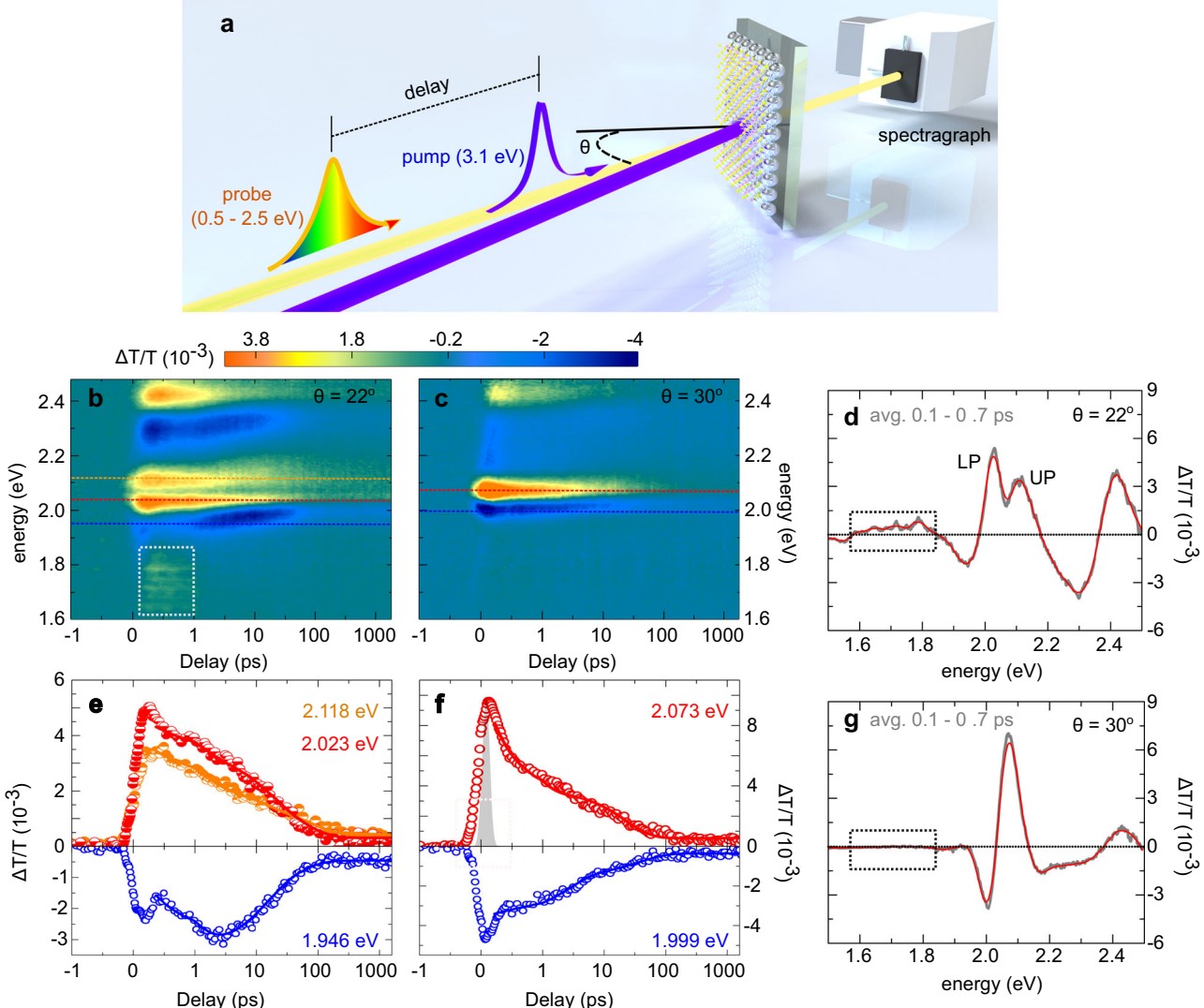

**Fig. 2 Transient optical responses of PC-WS$_2$ systems. a** Schematic of angle-resolved ultrafast pump-probe spectroscopy; (note: the pump and probe beams are nearly colinear in practical experiments) **b, d, e** Refer to normalised differential transmission spectra ($\Delta T/T$) at the tuned angle ($\theta = 22°$), while **c, f**, and **g** refer to $\Delta T/T$ at the detuned angle ($\theta = 30°$). **b, c** Intensity plots of $\Delta T/T$ as function of time delay and probe photon energy, using the same colour bar (which is also used by Fig. 3a). **d, g** $\Delta T/T$ spectra averaged within the time span from 0.1 to 0.7 ps after pump. **e, f** $\Delta T/T$ transient at specific energies (labelled with different colours), in which scatter symbols and solid curves represent measured and fitted data, respectively. Dashed frames in panel **b, d, g** Mark the spectral region of the broad maxima (see main text). All measurements were carried out using 400 nm ($E = 3.1$ eV) pump pulses that have 100 fs duration and pump fluence of 12 μJ cm$^{-2}$ at room temperature. The instrument-response-function is shown as the grey area in **f**. All time axes are displayed in logarithmic scales.

with $\Delta T/T$ spectra of bare $WS_2$ MLs (Supplementary Fig. 4), can be attributed to the relaxation of exciton A. However, no relaxation features from the detuned plasmons are observed. We also note that the plasmon features at the tuned frequency (2.05 eV) are absent in $\Delta T/T$ of bare PCs too (Supplementary Fig. 5). As a previous study[24] pointed out, this is the result of very ineffective excitation of plasmons under the off-resonance pump (3.1 eV). In other words, the lattice plasmons can only be effectively excited by coupling to excitons under current pump conditions.

What is interesting is that when the PC-$WS_2$ system is in tune (Fig. 2b), there appears a $\Delta T/T$ maximum lasting for ~1 ps at the frequency range from $E = 1.6$–1.8 eV, which, according to the integrated $\Delta T/T$ spectrum near zero probe delay (Fig. 2d), has positive magnitudes, indicating negative optical absorption or positive gain. In contrast, this feature is remarkably weaker in the detuned state (Fig. 2c, g) and is completely absent in bare $WS_2$ MLs (Supplementary Fig. 6) and the bare PC samples (Supplementary Fig. 5).

To confirm our observations, we have performed measurements under ~10 times higher pump fluence (100 µJ cm$^{-2}$) for the tuned polaritons (Fig. 3a), where the broad maxima become much more pronounced. In addition, we can see large spectral shift (Supplementary Fig. 15) as well as remarkably delayed occurance of UP and LP maxima (Fig. 3b and Supplementary Fig. 16), which indicate significant enhancement of the system's nonlinear optical responses (Details in Section 8 of SI). Similar to the low-power case, the transient variation of the broad maximum (Fig. 3c) takes ~1.5 ps from initial excitation to fading. Figure 3d shows the development of the broad maximum

upon pump fluence, where the magnitude and width of the maximum are highly dependent on pump intensity. Under 100 µJ cm$^{-2}$ pump fluence, the full-width at half-maximum reaches at ~200 meV with highly enhanced magnitudes as compared to the maximum under 5 µJ cm$^{-2}$ pump, also contrasting the flat spectral features in bare $WS_2$ MLs.

**Possible mechanisms of bandgap renormalisation.** According to the previous works[9,25,26], the broad maxima are a clear evidence of bandgap renormalisation accompanied by population inversion, which is broadly understood as a population of high-density carriers in semiconductor lattice. In particular, when semiconductors are pumped, the generated non-equilibrium photo-carriers will occupy electron and/or hole states, leading to the formation of new quasiparticle bandgaps, which is described by[27]:

$$\Delta E_g = -\sum_{q \neq 0} V_s(q)\,[f_e(q) + f_h(q)] - \sum_{q \neq 0} [V_s(q) - V(q)], \quad (1)$$

where $V_s(q)$ and $V(q)$ represent fourier transforms of screened and unscreened Coulomb potentials, while $f_e(q)$ and $f_h(q)$ are occupation probabilities of electron and hole with momentum $q$. The onset of the new bandgap can be extracted from the low-energy end of the broad maximum[9] (red dashed vertical line in Fig. 3c). It means that in our experiments, the renormalised bandgap starts at $E_g \approx 1.60$ eV, lying ~400 meV below LP and ~550 meV below the bandgap of $WS_2$ MLs (given that the binding energy of exciton A is decreased to ~100 meV when deposited on metal substrates[28], i.e., about a half of its initial value[19]).

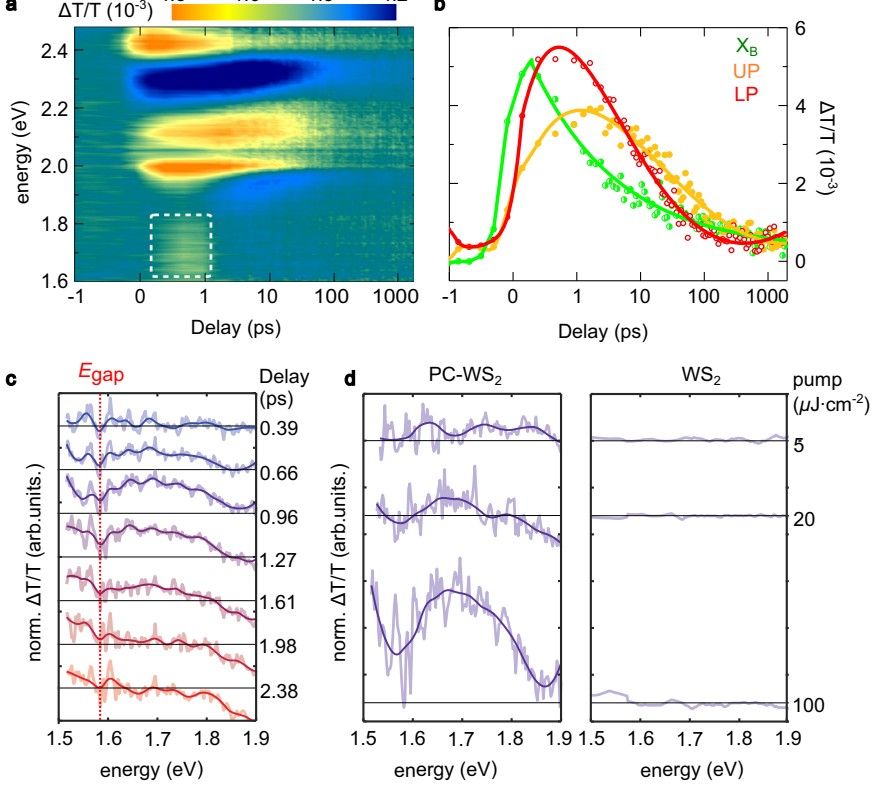

**Fig. 3 Transient optical responses under high-power pump. a** Intensity plot of $\Delta T/T$ spectra of PC-$WS_2$ under 100 µJ cm$^{-2}$ pump fluence at $\theta = 22°$, where orange (blue) colour represents the maximum (minimum) value. **b** Delay time dependent spectra ($\Delta T/T$) at energies of UP, LP, and exciton B extracted from panel **a**. Solid curves are plotted only for visual guidance. **c** $\Delta T/T$ spectra at different delay times, extracted from the white dashed frame in **a**; red dashed vertical line indicates the onset of renormalised bandgap. **d** comparison of $\Delta T/T$ spectra at delay of 0.96 ps between PC-$WS_2$ (left) and $WS_2$ MLs (right) under gradually increasing pump fluence; horizontal lines indicate the $\Delta T/T$ amplitudes of zero.

These results demonstrate a giant bandgap renormalisation in 2D semiconductors under such low pump fluence (down to ~10 μJ cm$^{-2}$). Specifically, this is three orders of magnitude lower than the photoexcitation (3400 μJ cm$^{-2}$ at room temperature) used in similar studies[9]. In their case, the ultrastrong pump enormously enhances exciton–exciton interactions in WS$_2$ single/bi-layers, reducing exciton binding energy, finally breaking excitons into unbound electron-hole plasma. This effect is known as Mott-transition, resulting in a high-density carrier population that leads to a large bandgap renormalisation[9,25].

In our system, the carrier density may be increased by plasmonic absorption enhancement (PAE), which, however, can not provide enough carrier population according to our calculation. Specifically, the excitation of plasmons can enhance the absorption of the pump by the system, which naturally results in an elevation of carrier numbers in the lattice. In the PC-WS$_2$ system, the pump intensity can be amplified at the position of the WS$_2$ ML (Supplementary Fig. 13), which, according to our calculation, gives a ~5 times average increase of absorption in the semiconductor. As a result, the carrier density can achieve up to ~1.2 × 10$^{13}$ cm$^{-2}$ if the absorbed pump energy is fully converted. However, even this overestimated value is still one order of magnitude lower than the density level (~10$^{14}$ cm$^{-2}$)[9] required to cause a Mott-transition at 70 K, let alone the level at room-temperature. Furthermore, PAE should also enhance carrier generation in the detuned systems. However, in our experiments, only the tuned system shows a large bandgap renormalisation (Fig. 2b, d). Hence, there must be other mechanisms that can enhance carrier population in addition to PAE.

It is also noted that the photoinduced absorption maxima (presenting as minima in negative $\Delta T/T$ magnitudes) in the tuned polaritons are clearly delayed as compared to its counterpart in the detuned polaritons (blue curves in Fig. 2e, f and blue dashed line indicated areas in Fig. 2b, c). In bare TMDC MLs[17–19,29], such postponed absorption maxima are usually observed under high-intensity pumps (e.g., Supplementary Fig. 6c and Supplementary Fig. 7), which, similar to Mott-transition, owes to enhanced exciton–exciton and/or exciton–electron interactions induced by high-power pump (see Section 3 and 7 in SI for detailed discussions). In contrast, the delayed maxima in our hybrid systems appear under much weaker pump and are only associated with the tuned polaritons. Together with the bandgap renormalisation, all these evidences suggest that the presence of additional carriers highly relate to the plasmon–exciton coupling.

**Coherent doping of plasmonic hot electrons**. In this case, another source is more likely to contribute sufficient carriers, i.e., hot electrons from the plasmonic crystal. Specifically, the excited plasmon modes may dephase from wave-like states through non-radiative decay, which generates electrons with non-equilibrium thermal distributions[13]. As a result, if plasmonic resonators are integrated with semiconductors, the non-equilibrium electrons that have energies higher than the charge barrier formed at the metal–semiconductor interface can enter the lattice through direct electron transfer (DET)[30,31], known as "hot electrons". As the barrier can prevent charges from returning back to the metals[30,32], hot electrons can dope adjacent semiconductors[33], modifying their photovoltaic and photocatalytic performance[12]. (See Section 4 in SI for details.)

The sign of hot electron doping has already been shown in the transient spectral features of our system. Specifically, the polaritons at the tuned state (Fig. 2e) show slower relaxations than those of the detuned state (Fig. 2f and Supplementary Table II) and uncoupled excitons (Supplementary Table I). This, according to a previous study[24], is the result of coupling between plasmons and semiconductor excitons. In particular, the generated plasmonic hot electrons can populate the semiconductor lattice accompanying the coherent energy exchange between plasmons and excitons. Since the exchange rate is very high (2$g_{1A}$ ≈ 174 meV), the hot electron population runs at an ultrashort period of ~25 fs ($T_R = 2\pi/2g_{1A}$)[34], which is greatly shorter than the exciton formation (<1 ps)[29], the non-radiative decay (at scales of 10 ps) and the radiative decay process (up to few-hundred ps) in WS$_2$ MLs[17,18]. It means that during polariton relaxation, hot electrons can repeatedly fill the unoccupied states in the WS$_2$ conduction band, which slows down the exciton relaxation via Pauli blocking, leading to extended lifetimes[24].

**Calculation of hot electron density**. To prove that hot electron doping can induce the observed bandgap renormalisation, we need to understand how the hot electrons are generated in our system as well as quantify the net carrier density in the lattice. As explained before, due to the off-resonance frequency of the pump, plasmons in the PC-WS$_2$ system can only be effectively excited by coupling to excitons. Specifically the pump energy is absorbed by the semiconductor and down-converted to excite the plasmon–exciton polaritons, which, as half-plasmon half-exciton hybrid states, naturally excite their plasmonic component and result in the generation of plasmonic hot electrons. These charges then overcome the tunnelling barrier ($\Delta\phi_{TB}$) formed at the Ag–Al$_2$O$_3$–WS$_2$ interface to dope the WS$_2$ lattice. During this process, the hot electron doping is subject to several major losses, including (i) the limited pump absorption by the WS$_2$ MLs, (ii) the losses in energy down-conversion and (iii) the losses due to the hybrid nature of polaritons.

Following this process, we have developed a model to numerically estimate the density ($N_e$) of hot electrons doped in the WS$_2$ lattice, which starts with the Hamiltonians of an electron Fermi gas and is then involved with plasmonic excitations[35] and losses in the conversion process:

$$N_e = \frac{F_{pump} \cdot \eta_A \cdot \eta_D \cdot \eta_{pl}}{2c\epsilon_0} \cdot \mathcal{F} \cdot \frac{1}{\pi^2} \frac{e^2 E_F^2}{\hbar} \frac{\hbar\omega - \Delta\phi_{TB}}{(\hbar\omega)^4}, \quad (2)$$

where $e$ is the electron charge, $\omega$ is the frequency, $E_F$ is the Ag Fermi-level, $c$ is the light of speed, $\epsilon_0$ is the vacuum permittivity and $F_{pump} = 12$ μJ cm$^{-2}$ is the pump fluence. Here we take $\eta_A \approx 55\%$ as the absorption coefficient, $\eta_D \approx 66\%$ as the energy down-conversion ratio and $\eta_{pl} \approx 50\%$ as the excitation ratio of the plasmon component in polaritons. As a result, $F_{pump} \eta_A \eta_D \eta_{pl}$ corresponds to the process that the pump energy is absorbed, down-converted and coupled to the plasmonic components in polaritons with major losses included. As optical modes, the excited polaritons gain a spatial distribution (Fig. 4b) at the tuned frequency, spreading over the Ag cap surface with hot spots at the interstices between caps. This can be mathmatically expressed as $\mathcal{F} = |E/E_0|^2$, where $E_0$ refers to the incident field and $E$ stands for the enhanced field. In addition, the tunnelling barrier $\Delta\phi_{TB}$ is set to be 1 eV, which is a typical value for the ultrathin Al$_2$O$_3$ layers used in our system[31,36], and this setting can help address other dissipations that are not considered in the whole excitation process. (See Section 6 in SI for detailed discussions about the model development and parameters taken.)

Using Eq. (2), we are able to plot the spatially distributed hot electron density in the WS$_2$ monolayer (Fig. 4c). The density naturally acquires identical distributions as do the plasmonic excitations, exhibiting inhomogeneous distribution over the area. It has values typically higher than $1 \times 10^{13}$ cm$^{-2}$ in most of the areas, peaking at the interstices between caps with maxima larger than $2 \times 10^{14}$ cm$^{-2}$. We want to point out that these heavily doped areas are the parts that can be optically detected in

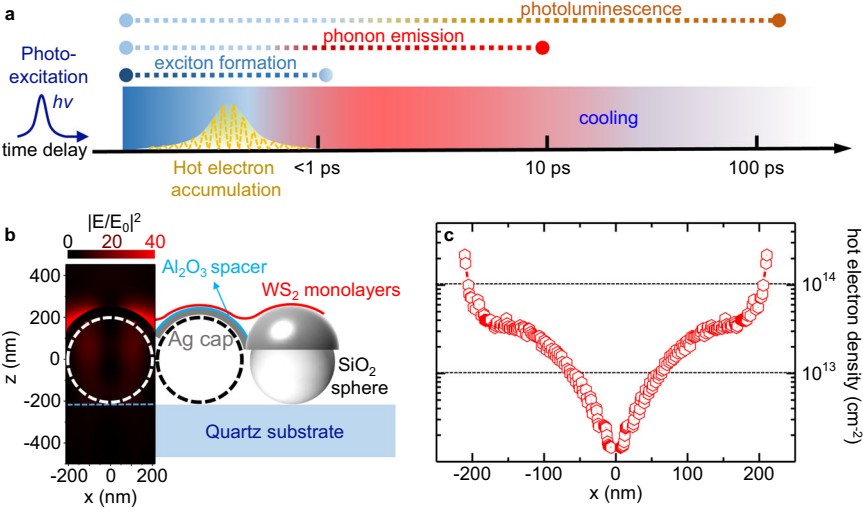

**Fig. 4 Hot electron doping and density. a** Schematic illustration of the overall relaxation scenario after pulsed optical excitation on the PC-WS$_2$ systems, where the yellow oscillating dashed curve stands for repeated hot electron population in lattice; dotted lines above correspond to the time durations of different relaxation processes; $h\nu$ refers to energy of the pump pulse; **b** simulated intensity distribution $\left|\mathbf{E}/\mathbf{E}_0\right|^2$ in a single sphere-cap unit (XZ cross-section at middle of the sphere) at the frequency of 2.05 eV, together with a schematic of the PC-WS$_2$ sample; please note that the WS$_2$ ML is partially suspended at interstices between two Ag caps, but not fully covering the metal surface; detailed morphology can be referred to our previous work[15]; **c** calculated hot electron density along the curved WS$_2$ surface as a function of distance along the projected x-direction (at the same cross-section as in **b**).

transient experiments. As discussed in our previous works[14,37], the far-field optical signals directly relate to the near-field distributions. As the result, the detected spectral features mainly result from the photons that are radiated from the hot spots, but not other locations in the PC-WS$_2$ system. Therefore relevant spectral features should be analysed on the basis of spatially distributed carrier density but not the averaged one.

## Discussion

We note that with major losses taken into account, our model provides a relatively accurate estimation of carrier density in the WS$_2$ lattice. Specifically, the development of a complete Mott-transition in a WS$_2$ monolayer requires a carrier density up to $\sim 10^{14}$ cm$^{-2}$[9]. However, the injected carriers in our system are hot electrons, which are different from the dessociated electron-hole pairs induced by pure optical pumping[9,26], but are more similar to free charge carriers by electrical injection[8,38], where a $\sim$550 meV bandgap renormalisation in WS$_2$ MLs can occur at the electron density of $3 \times 10^{13}$ to $1.1 \times 10^{14}$ cm$^{-2}$[8]. These results share high similarity with our observation, suggesting that the hot electron doping in our system is able to achieve the threshold to induce a bandgap redshift up to $\sim$500 meV with carriers draining from conduction band K to $\Sigma$ valley[39] that renders the semiconductor indirect.

Given that there is little evidence for other possible carrier sources, e.g., polariton condensates[40], we conclude that coherent doping of plasmonic hot electrons is dominantly responsible for the spectral and transient features that require high-density population. In particular, the hot electron population starts with polariton formation and repeatedly takes place throughout the whole relaxation process. Owing to the existence of the tunnelling barrier, hot electrons can be accumulated in the lattice before they decay (within 1 ps[13]), which simultaneously competes with rapid exciton relaxations, transiently converting the intrinsic WS$_2$ monolayers to "n-doped" ones. This leads to the giant bandgap renormalisation with population inversion that peaks at few-hundred femtoseconds (Figs. 2b and 3c), and also induces the delayed maxima in Fig. 2b, e. When we enhance pump power, the

hot electron density is accordingly increased, even capable of delaying the occurance of excition relaxation (Fig. 3b and Supplementary Fig. 16).

As discussed above, the intermediate plasmon–exciton coupling dramatically modifies the electronic band structures of WS$_2$ monolayers, which is induced, to a large degree, by plasmonic hot electron doping via coherent plasmon–exciton coupling. This effect can hardly be observed in traditional exciton-polaritons[40], being a non-trivial factor that has to be considered when studying light-matter interactions using plasmonic resonators, which, on the other hand, provides effective measures to engineer bandgap of 2D semiconductors and benefit relevant applications[41,42].

## Methods

**Sample preparation.** Plasmonic crystals were prepared by self-assembly techniques together with thermal evaporation. In particular, silica spheres with diameter of 425 nm were self-assembled to a hexagonally packed monolayer on quartz substrates using a reported method[14], which is followed consecutively by evaporation of 40 ± 5 nm thick silver film and 2.5 ± 2 nm thick Al$_2$O$_3$ film. The metal and dielectric evaporation acquires the shape of semi-spheres, resulting in an array of silver caps with a hexagonal lattice. The thickness of the metal and dielectric layers are characterised using an ellipsometer measuring identical evaporations on flat silicon substrates.

WS$_2$ monolayers were prepared using chemical vapour evaporation with sulfur powder and tungsten dioxide powder being precursors, respectively. Using the poly (methyl methacrylate)-assisted transfer method[43], we can transfer the prepared WS$_2$ monolayer onto the PC surface. It is noted that the presence of WS$_2$ monolayers is hardly seen from SEM images of our sample due to the ultrathin thickness of the monolayer (<1 nm). For more details about the fabrication and the monolayer morphology of the PC-WS$_2$ samples please see our previous publication[15].

**Optical characterisation.** The large sample areas (2–4 cm$^2$) allow us to conveniently perform femtosecond pump-probe absorption measurements[44] with different incident angles. Specifically, as shown in Fig. 2a, the PC-WS$_2$ samples were excited by a pump pulse, followed by a delayed broadband illumination to probe the photoexcitation induced changes, which can be characterised by measuring the differential transmission, defined as the normalised change of the probe transmission induced by the pump $\Delta T/T = (T - T_0)/T$, at different delayed times. The spectrometer in the pump-probe set-up is composed of a camera and a spectrograph, while the spectrograph is a home-built device using a prism to disperse the incident probe beam. The pump pulse has a frequency of 3.1 eV

(400 nm), much higher than exciton A (~2.05 eV), while the probe pulse has a broadband frequency from 0.5 to 2.5 eV. The whole ultrafast system acquires a pulse resolution of 100 fs, and were converted to p-polarisation (electric field parallel to the plane of incidence) before entering samples. The transmission spectra of PC-WS$_2$ samples in equilibrium states (without photoexcitation) were preformed using the reported method[15]. All optical characterisations were carried out at room temperature.

## Data availability

All data present in the main text and/or the Supplementary Information are avaiable from the authors upon request.

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

## Acknowledgements

The authors acknowledge the New Idea Research Funding 2018 (Dodd-Walls Centre for photonic and quantum technologies), the Marsden Fast-start Fund by Royal Society of New Zealand through contract MFP-UOO1827 and MFP-VUW1715 and the Smart Ideas Fund by Ministry of Business, Innovation and Employment, New Zealand through contract UOOX1802. In addition, this work was supported in part by the National Key Research and Development Program of China (No. 2017YFA0205700) and the National Natural Science Foundation of China (Nos. 6192782 and 51861135201), Beijing Muni-cipal Natural Science Foundation (1214027), the Science and Technology Innovation Project of Beijing Institute of Technology and the NSFC funding (12004313). The authors also acknowledge the visiting Fellowship awarded by New Zealand Centre at Peking University. We thank Dr. M. Yan and Dr. F. Hong for their help with thin-film deposition, AFM, and SEM measurements. The authors appreciate valueable discussions with Dr. Michael Price.

## Author contributions

B.D. and Y.-H.C. conceived the project; Z.Z. and B.D. prepared the samples; R.T., K.C., Y.-H.C., F.L. and B.D. carried out the optical and other characterization; Y.-H.C. and B.D. performed the simulation; Y.Z., M.Q., R.J.B., J.M.H. and B.D. supervised the projects; B.D. prepared the manuscript; all authors discussed and analyzed the results.

## Competing interests

The authors declare no competing interests.
