## [Peer Review File · Nature Communications]

REVIEWER COMMENTS

Reviewer #1 (Remarks to the Author):

The authors report an interesting study on doping two-dimensional semiconductors using plasmonic hot electrons. Specifically, the proposed plasmonic crystal (PC)- tungsten disulfide (WS₂) structure is investigated extensively with transient absorption (TA) spectroscopy. While this study provides a unique way to engineer the optoelectronic properties of 2D semiconductors, there are some issues that need to be resolved as listed below:

1. The main conclusion of this work is that the bandgap of WS₂ can be significantly modified by hot electrons injected from the PC. However, there is no strong evidence that hot electrons dominate the observed transient features in TA spectroscopy:

1.1. According to the discussions, hot electrons are believed to be injected into WS₂ by tunneling through the Al₂O₃ spacer layer. While the spacer layer helps to prevent injected hot electrons from tunneling back into the PC, it also decreases the hot electron injection efficiency and this effect depends on the spacer layer thickness. It is not clear how the authors determine the thickness of the Al₂O₃ to use for this work and if this thickness is favorable for DET.

1.2. Similarly, the authors need to explain why the tunneling barrier height can be set to be 1 eV in Equation (2).

1.3. Since losses are not included in Equation (2), the actual number of injected hot electrons could be much lower than calculated. The authors should provide a better evaluation of the effect of losses either theoretically or experimentally.

2. Since the quality of the silver cap layer can affect the generated hot electrons and local electric field distribution, did the authors check the optical properties of the deposited silver layer? If so, they should show results of the optical measurements. Otherwise, those measurements need to be made for the next revision.

3. To compare the broad maxima in Figure 2d and 2g, it would be more helpful to show both figures with the same y-scale.

4. The authors also mentioned the enhanced nonlinear optical responses of the PC-WS₂ structure. How is it related to the rest of this work? The authors should elaborate more on the relevance of the enhanced nonlinear response.

I would recommend a major revision for the authors to address the questions above.

Reviewer #2 (Remarks to the Author):

The authors study nonlinear optical properties of plasmonic crystals covered with WS₂ monolayers using time-resolved transient absorption measurements. The time resolution is of the order of 100 fs, insufficient to resolve coherent exciton-plasmon coupling phenomena in the time domain. The samples show some signatures of exciton-plasmon coupling. Their linear optical properties have been studied in earlier work.

The authors do claim that the samples are in the strong coupling regime but, in my opinion, this is actually not convincingly supported by the data shown in the paper or in the SI. It seems to me that all the spectra that are shown are consistent with an exciton-plasmon system in the intermediate coupling regime (exciton lifetime < rabi period < plasmon lifetime). Support for the strong coupling claim in the paper should be given by a quantitative analysis of the line shape of the spectra. This criticism, however, is only partially relevant for the main results of the present paper probing incoherent optical nonlinearities occurring on time scales that are longer than the lifetimes of coherent excitations in the coupled system. The main experimental observations are potentially of interest even when the strong-coupling claim is withdrawn.

In Fig. 2, the authors show angle-resolved studies of the nonlinear optical response of their system. Several interesting features are observed. First of all, some spectra display a transient below-bandgap response, similar to what has been shown by Chernikov et al. for bare WS₂ layers in Nature Phot. 9, 466 (2015). Second, the authors see a strongly angle-dependent differential transmission lineshape around the WS₂ A exciton at 2.1 eV.

Let us start with the second feature – which is not discussed at all in the paper. What is the origin of the lineshapes in Fig. 2d and g? What explains the angle-dependent change in lineshape seen in Fig. S12? If the excitons and plasmons in the present system are indeed coupled, it should be possible to analyze these lineshapes in terms of a phenomenological coupled Lorentz oscillator model as routinely used for describing such coupled systems. I would like to strongly encourage the authors to perform such an analysis. It would provide interesting new insight into the nonlinear optical properties of TMDC excitons coupled to plasmons. Also, it could certainly help the authors to better understand some of their results. Based on such analysis, a more quantitative and less speculative analysis of the dynamical behavior should be possible.

Now, let us turn to the below-bandgap absorption peak. The spectral signatures are indeed very similar to what has been seen by Chernikov in his Nature Photonics paper. I therefore did not quite understand why the authors argue that their results “sharply contrast” with those earlier observations. I would guess that the nature of the below-bandgap peak is very much the same as demonstrated earlier: bandgap renormalization due to a nonequilibrium carrier concentration in WS₂ film. The fact that this is seen here at lower pump powers seems not particularly surprising to me: the presence of the plasmonic crystal enhances the absorption (even at the off-resonant pump energy). Since I agree with the authors that the data show this below-bandgap peak at lower fluence than observed before, I believe that the findings are interesting. I would therefore like to ask the authors to thoroughly discuss their observations and to comparatively discuss different physical mechanisms that could account for it. In the present manuscript, they focus very much on a “hot electron from plasmonic crystals” explanation which – in my opinion – is not necessarily supported by the data. I think that it is realistic to estimate plasmonic absorption enhancements at the pump energy and to carefully compare photoinduced carrier densities with and without plasmonic field enhancement. Can plasmonic field enhancement account already for the increase in bandgap renormalization? This question should be discussed in much more depth than in the present manuscript and – if possible – supported by simulations of the linear optical properties of the sample.

More technical issues:

1. As mentioned already, I do not believe that the claim of “fast and repeated hot electron population” (aka strong coupling) on page 8 is justified.
2. A color code in Fig. 1b is missing.
3. A color code in Fig. 3a is missing.
4. Why are the data in Fig. 3c, d so noisy. The time-dependent spectra in Fig. 3a look reasonably clean. Has Fourier filtering and smoothing been applied to them.
5. The physics behind Eq. 2 remains unclear since the model is not explained in the text. If the authors want to maintain their “hot electron” explanation, the model needs more thorough explanation.

In summary, the reported show some new aspects of the transient optical nonlinearities of exciton-plasmon-coupled systems and hence may be of interest to the research community. In order to make the data publishable a much more in-depth analysis and discussion is needed. Claims of “strong coupling” should either be proven or withdrawn.

Christoph Lienau

Reviewer #3 (Remarks to the Author):

The authors reported optical control of bandgap of a monolayer WS₂ integrated with a plasmonic structure. The sample studied contains a monolayer WS₂ on a self-assembled plasmonic crystal. By coupling the plasmonic resonance with A-excitons of WS₂, the authors demonstrate control of the WS₂ bandgap by optically exciting plasmonic hot electrons that transfer to WS₂. Ultrafast control of bandgap renormalization can have important applications in ultrafast photonics such as ultrafast optical switch. As such, the results are significant. However, there are a few issues that I wish the authors could clarify.

1. The main conclusion of the study is a 650-meV bandgap renormalization at RT. The authors need to provide strong evidence and more detailed analysis to support this claim. It appears that the authors assumed an exciton binding energy of 200 meV. However, this is the value for a WS₂ monolayer on an insulating substrate. When coupled with a plasmonic crystal, this is likely to change, since such a large exciton binding energy is due to the lack of screening in the 2D form. Without knowing the precise value of the exciton binding energy in their structure, the claimed renormalization may not be accurate.

2. It is also a bit concerning to use the peaks in TA spectra as the parameter to determine the bandgap. The TA peak aligns with the resonance only if the phase-space filling is the dominant mechanisms of TA. In 2D materials, this may not be the case.

3. The authors compared with Ref. 9 in terms of the achieved normalization and the used injection level. However, the hot-electron inject method used here results in net charges in WS₂, which is quite different from optical doping. The authors should discuss about the potential implications. In addition, a direct comparison of the pump fluence used may not be fair, since the absorptions are different.

AUTHOR'S RESPONSE TO REVIEWERS

Reviewer #1

We are glad that Reviewer #1 thinks that *“The authors report an interesting study on doping two-dimensional semiconductors using plasmonic hot electrons.”*. In addition, the reviewer also suggests that:

1. We should provide stronger evidence that hot electrons dominate the observed transient features by
1.1 providing more details on the determination of the thickness of the Al₂O₃ spacer.

Author response: We very much thank the reviewer and agree with her/him that the spacer thickness is critical in forming a favourable tunneling barrier for hot electron transfer. We therefore have added the following sentence in the main text (line 246ff):

“.....The thickness of the metal and dielectric layers are characterised using an ellipsometer measuring identical evaporations on flat silicon substrates.”

In addition, we have also added a schematic in Fig.S14 in the Supplementary Information (SI) to show the surface morphology of Al₂O₃ layer. Relevant discussion can be found in the response to the below suggestion.

T.2 We should explain why the tunneling barrier height can be set to be 1 eV in Eq.(2) in the main text and the relation between the barrier height and the Al₂O₃ spacer thickness.

Author response: We appreciate that Reivewer#1 gives us this opportunity to better explain the height of tunneling barrier. As the reviewer suggested, the barrier height closely relates to the spacer thickness. In our sample, the spacer thickness varies with different locations at a level of 2.5 ± 2 nm. According to the previous work [*Nanoscale*, **11**, 4811, (2019)], the Al₂O₃ spacer with a thickness of ~ 2.1 nm can form a tunneling barrier of ~ 0.8 eV at the metal-WS₂ interface. Therefore here we take $\Delta\phi_{TB} = 1$ eV, which is also a value that is commonly used in other studies, e.g. [*Adv. Opt. Mater.*, **5**, 1600594, (2017)] and [*ACS Photonics*, **4**, 2759, (2017)]. To clearly elucidate the value setting, we have added a paragraph (line 419ff) and Fig.S14 in the SI and the following sentence (line 195ff) in the main text:

“....In addition, the tunneling barrier $\Delta\phi_{TB}$ is set to be 1 eV, which is a typical for the ultrathin Al₂O₃ layers used in our system[31, 36], and this setting can help address other dissipations that are not considered in the whole excitation process....”

with references:

[31] Xiang Tian Kong, Zhiming Wang, and Alexander O. Govorov. “Plasmonic Nanostars with Hot Spots for Efficient Generation of Hot Electrons under Solar Illumination”, *Adv. Opt. Mater.*, **5**, 1600594, 2017.

[36] Shan Zheng, Haichang Lu, Huan Liu, Dameng Liu, and John Robertson. “Insertion of an ultrathin Al₂O₃ interfacial layer for Schottky barrier height reduction in WS₂ field-effect transistors”, *Nanoscale*, **11**, 4811, 2019.

1.3 Reviewer #1 suggests that we should provide a better evaluation of the losses for calculation of hot electron density using Eq.(2).

Author response: We are very much grateful that the reviewer has raised this suggestion and agree with the reviewer that it is necessary to include losses in the calculation of hot electron density, as the density is the critical factor in identifying the role of hot electrons in the observed bandgap renormalisation. We have therefore revised Eq.(2) to a new form:

$$N_e = \frac{F_{\text{pump}} \cdot \eta_A \cdot \eta_D \cdot \eta_{\text{pl}}}{2c\epsilon_0} \cdot \mathbf{F} \cdot \frac{1}{\pi^2} \frac{e^2 E_F^2 \hbar\omega - \Delta\phi_{\text{TB}}}{\hbar (\hbar\omega)^4} \quad (2)$$

In short, three major losses have been included: (i) $\eta_A = 55\%$ is the ratio of pump energy that is absorbed by the system; (ii) $\eta_D = 2/3$ presents the down-converted energy ratio to excite polaritons; and $\eta_{\text{pl}} = 50\%$ characterises the ratio of plasmonic component in polaritons. The detailed discussions of these major losses and the deduction of the revised Eq.(2) have been added into both the SI and the main text. Specifically, a new sub-section “**Including losses**” has been added in the Section 6 of the SI, which includes Fig.S13 and relevant discussions. In addition, the following sentence has been added into the main text (line 173ff):

“To prove that hot electron doping can induce the observed bandgap renormalisation, we need to figure out how the hot electrons are generated in our system as well as quantify the net carrier density in the lattice. As explained before, due to the off-resonance frequency of the pump, plasmons in the PC-WS₂ system can only be effectively excited by coupling to excitons. Specifically the pump energy is absorbed by the semiconductor and down-converted to excite the plasmon-exciton polaritons, which, as half-plasmon half-exciton hybrid states, naturally excite their plasmonic component and result in the generation of plasmonic hot electrons. These charges then overcome the tunneling barrier ($\Delta\phi_{\text{TB}}$) formed at the Ag-Al₂O₃-WS₂ interface to dope the WS₂ lattice. During this process, the hot electron doping is subject to several major losses, including (i) the limited pump absorption by the WS₂ MLs, (ii) the losses in energy down-conversion and (iii) the losses due to the hybrid nature of polaritons.”

and in line 188ff in the main text:

“...Here we take $\eta_A \approx 55\%$ as the absorption coefficient, $\eta_D \approx 66\%$ as the energy down-conversion ratio and $\eta_{\text{pl}} \approx 50\%$ as the excitation ratio of the plasmon component in polaritons. As a result, $F_{\text{pump}} \cdot \eta_A \cdot \eta_D \cdot \eta_{\text{pl}}$ corresponds to the process that the pump energy is absorbed, down-converted and coupled to the plasmonic components in polaritons with major losses included...”

It turns out that with losses included, the hot electron density in the WS₂ monolayer can still typically achieve $\sim 10^{13} \text{ cm}^{-2}$, even approaching 10^{14} cm^{-2} at hot spots. These numbers have reached what is required ($3 \times 10^{13} - 1.1 \times 10^{14} \text{ cm}^{-2}$) to induce a $\sim 550 \text{ meV}$ bandgap renormalisation by electrical doping, meaning that the hot electron densities are sufficient to result in the observed bandgap restructuring. For more details, please see the revised **Fig.4** and relevant paragraphs in the main text (lines 200 - 219 ff).

2. Reviewer #1 suggests that we demonstrate the optical properties of the bare plasmonic crystal (PC) in the manuscript, since the quality of the silver PC can affect the generated hot electrons and local electric field distribution.

Author response: We agree with the reviewer. Due to the frame limit, we have included the angle-

resolved transmission spectra of the bare PC sample in Fig.S1(a) of the SI for comparison with the spectra of the PC sample integrated with a WS₂ monolayer [PC-WS₂, Fig.S1(b)].

3. Reviewer #1 suggests that we show Fig.2d and Fig.2g with the same y-scale, which will provide a better comparison for the bandgap renormalisation effect.

Author response: We thank the reviewer for her/his careful observation. We therefore have made relevant revision to Fig.2d and 2g in the main text.

4. Reviewer #1 suggests that we elaborate more on the relevance of the enhanced nonlinear response in our system (Fig.3a and 3b in the main text, and Fig.S15 and S16 in the SI).

Author response: We very much thank Reviewer #1 for this valuable suggestion, since this has inspired us to explore a new research direction. The nonlinear response of our system under high-power pump mainly includes: (i) the spectral shift (Fig.S15) and (ii) the delayed occurrence (Fig.S16) of the polariton maxima.

In short, by comparing the relative magnitudes of the polariton split maxima, we find that the spectral shift highly relates to the excitonic resonance shift that is induced by carrier density enhancement. Likewise, the delayed occurrence of maxima can also relate to the enhancement of carrier density in the lattice. These results correspond to the main conclusion of this manuscript, i.e. the strong coupling between plasmons and excitons facilitates the generation of hot electrons, resulting in the elevation of carrier density in the lattice. We have added these discussions into Section 8 of the SI.

Reviewer #2

We are very grateful that Reviewer #2 positively commented on our results, e.g. *“The main experimental observations are potentially of interest”*, *“I believe that the findings are interesting.”* and *“the reported show some new aspects of the transient optical nonlinearities of exciton plasmon- coupled systems and hence may be of interest to the research community”*. In addition, s/he has provided many valuable suggestions:

1. Reviewer #2 strongly encourages that we perform lineshape analysis on spectral features of our system using a phenomenological coupled Lorentz oscillator model, since this enables more quantitative and less speculative analysis on the coupling state of the system, providing a solid ground for other discussions in the manuscript.

Author response: We fully agree with Reviewer #2 and very much appreciate this suggestion, as it allows us to gain a much better understanding of the coupling behaviours in our system. Following the her/his instructions, we have used a coupled Lorentz model to fit the steady-state transmission spectra of both the bare PC and PC-WS₂ systems at different angles, which can be seen from Fig.S2 in the SI, where only the spectra at $\theta = 22^\circ$ are shown.

It turns out that the rate of energy exchange between plasmons and excitons ($2g$) is slower than the plasmon dephasing (κ) but faster than the exciton decay rate (γ), i.e. $\kappa > 2g > \gamma$, but the coupling strength in our system can still achieve $2g > (\kappa + \gamma)/2$, which is typically treated as in the strong coupling regime, according to an extensively used criterion in coupling systems, e.g. in the works [Khitrova et al., *Nat. Phys.*, **2**, 81 (2006)] and [Lienau et al., *ACS Nano*, **8**, 1056 (2014)].

Figure.S2 (SI): Analysis of spectral line shape and plasmon-exciton coupling.

We want to point out that the criterion of strong coupling highly depends on specific scenarios. For example, in traditional studies of cavity quantum electrodynamics, a vacuum Rabi splitting that is smaller than the cavity loss may result in unwanted decoherence among excitations. Therefore it requires a strict coupling criterion. But for some other situations, like in our system, the strong coupling effect is only used to provide a channel for energy transfer between plasmons and excitons. In this case, the relatively loose coupling criterion [$2g > (\kappa + \gamma)/2$] may apply, because the plasmon-exciton hybrid state can still remain effective in the presence of moderate cavity losses.

Figure.S3 (SI): Mixing coefficients in the strong plasmon-exciton coupling.

The effectiveness of the hybrid state can be determined through the relative degree of mixing between the cavity-plasmon and the exciton in plasmon-exciton polaritons. In short, following the method from [Lienau et al., *ACS Nano*, **8**, 1056 (2014)], we have included the damping factors (e.g. $i\kappa$ and $i\gamma$), using complex frequency of plasmons, excitons and polaritons to calculate the mixing coefficients for both upper and lower polaritons. As shown in the Fig.S3(b) of SI, it turns out that the coefficients calculated from experimentally acquired parameters slightly drift from the damping-free curves, but still reach $\sim 50\%$ at the tuned state. This means that the plasmonic component and excitonic component account for half of the polariton energy respectively, indicating that moderate losses in plasmons do not significantly change the strong coupling nature of the PC-WS₂ system in terms of energy exchange. Theoretical simulations also support this conclusion [Fig.S3(a)].

Therefore, here we kindly ask the reviewer that if s/he can consider the situation and allow us to treat the system as in the strong coupling regime. To better demonstrate the coupling behaviours in our system, we have rewritten the whole Section 1 in the SI, including newly added figures Fig.S1, S2 and S3. We have also added the following sentence in the main text (line 66ff):

“The splitting at the tuned state ($\theta = 22^\circ$) can be characterised by a vacuum Rabi splitting of $\hbar \cdot \Omega_R = \hbar \cdot 2g \approx 140$ meV, exceeding the widely used coherent strong coupling criterion $2g > (\kappa + \gamma)/2$ [21,22], where κ is the dissipation of plasmon modes and γ is the exciton decay rate.”

with references:

“[21] Khitrova, G., Gibbs, H. M., Kira, M., Koch, S. W. and Scherer, A. Vacuum Rabi splitting in semiconductors., Nat. Phys., 2, 81, 2006.”

“[22] Wang, W., Vasa, P., Pomraenke, R., Vogelgesang, R., De Sio, A., Sommer, E., Maiuri, M., Manzoni, C., Cerullo, G. and Lienau, C., Interplay between strong coupling and radiative damping of excitons and surface plasmon polaritons in hybrid nanostructures, ACS Nano, 8, 1056, 2014.”

In addition, we would like to point out that here all lineshape analyses were based on the steady-state transmission spectra, because the steady-state spectra provide stable, collective and power (and time) independent spectral features, which can simplify the discussion and is typically used for the analysis of plasmon-exciton coupling. In contrast, the spectral positions and linewidth of features in transient spectra $\Delta T/T$ are highly sensitive to the pump power and delay time (see Fig.2 and 3 in the main text and Fig.S4, S6, S8, S15, S16 and S17 and relevant discussions in the SI), which brings enoumours difficulties and uncertainties in analysing the coupling.

2. Reviewer #2 suggests that we discuss the *“strongly angle-dependent differential transmission lineshape around the WS₂ exciton A”*. Specifically, it is necessary to figure out *“the origin of the lineshapes in Fig.2d and 2g”* as well as to explain *“the angle-dependent change in lineshape seen in Fig.S12”*. The reviewer suggests that we use a Lorentz oscillator model to analyse the lineshapes.

Author response: We agree with the reviewer that it is important to discuss the lineshape change in transient differential spectra at distinct incident angles, as this will provide us a deeper understanding of the coupled system’s transient dynamics.

First of all, we want to point out that the coupled Lorentz oscillator model may not be applicable for analysing the lineshapes of differential spectra. Specifically, Lorentz models can describe relatively simple transition processes, e.g. excitations from ground to excited state or decay from excited to ground state, which, therefore, are typically used to analyse the simple resonant features in steady-state spectra. However, the differential spectra used here is $(T - T_0)/T_0$, which is the normalised subtractions between two spectra at different delay times, comprising of very complicated physical processes in addition to energy transition, e.g. the time-dependent many-body interactions between excitations. As a result, resonance features may broaden and shift at different delay times, leading to negative magnitudes and peak shifts as compared to the steady-state spectra. In this case, using Lorentz oscillator models to analyse the lineshapes of the differential spectra does not provide effective information.

Instead, we have phenomelogically analysed the lineshape change of differential spectra at different

angles and delay times, mainly drawing the following conclusions:

- (i) The splitting in differential spectra correspond to the splitting in the steady-state spectra, but with some frequency shifts and smaller splitting magnitudes.
- (ii) The frequency shifts and smaller splitting magnitudes should be the result of lineshape broadening and shift induced by many-body Auger recombination.

These discussions have been added to Section 2 in the SI, particularly with panel (d), (e) and (f) of Fig.S4 (Fig.S12 in the last version of SI)

3. Reviewer #2 suggests that we “*thoroughly discuss their observations and to comparatively discuss different physical mechanisms that could account for it.*” Specifically, the reviewer would like us to “*estimate plasmonic absorption enhancements at the pump energy and to carefully compare photoinduced carrier densities with and without plasmonic field enhancement.*” This should be “*discussed in much more depth than in the present manuscript*” and “*if possible – supported by simulations of the linear optical properties of the sample.*”

Figure.S13 (SI): Light absorption in PC-WS₂ system at the pump frequency (3.1 eV)

Author response: First we would like to express our sincere appreciation to Reviewer #2, since this suggestion has extended our understanding of carrier population in the system and greatly helped us reshape the later discussions on hot electron doping.

Specifically, we have modelled the intensity distribution of the PC-WS₂ system at the pump frequency [Fig. S13 (a)]. It turns out that the intensity enhancement at this frequency is no larger than ~ 7 times at the position of the WS₂ monolayer, which, according to our calculation, results in an average (over the whole area) optical absorption of $\sim 55\%$ in the monolayer. It means that the absorption in WS₂ is highly enhanced as compared to the bare monolayer ($\sim 10\%$) that is not integrated with the plasmonic crystal [Fig. S13(b)]. As a result, if the pump pulse that has a fluence of $12 \mu\text{J}/\text{cm}^2$ can be absorbed and fully converted, the carrier population in the WS₂ ML can reach at a density of $\sim 1.2 \times 10^{13} \text{cm}^{-2}$.

This overestimated value, however, is still one order of magnitude lower than the one ($\sim 10^{14} \text{cm}^{-2}$) that is required to develop a Mott-transition at a cryogenic temperature 70 K [Heinz et al., *Nat. Photon.*, **9**, 466, 2015], let alone the level at room-temperature. We also want to point out that even if all the absorbed energy in the PC-WS₂ system (i.e. $\sim 80\%$ absorption of pump, see the absorption spec-

trum of the PC-WS₂ at 3.1 eV in Fig.S13) can be converted to excite excitons in the WS₂ lattice, the generated carrier density is still lower than what is required to induce a large bandgap renormalisation as in our experiment.

In addition, the density increase induced by absorption enhancement should appear at transient spectra of all incident angles, but in our case, only the tuned state ($\theta = 22^\circ$) show a large bandgap renormalisation. Therefore, we do not think plasmonic absorption enhancement is the main factor that can induce the observed bandgap renormalisation in our experiments.

To present readers with this argument, we have added a new paragraph in the main text (line 129ff):

“In our system, the carrier density may be increased by plasmonic absorption enhancement (PAE), which, however, can not provide enough carrier population according to our calculation. Specifically, the excitation of plasmons can enhance the absorption of the pump by the system, which naturally results in an elevation of carrier numbers in the lattice. In the PC-WS₂ system, the pump intensity can be amplified at the position of the WS₂ ML (Fig.S13 in SI), which, according to our calculation, gives a ~ 5 times average increase of absorption in the semiconductor. As a result, the carrier density can achieve up to $\sim 1.2 \times 10^{13} \text{ cm}^{-2}$ if the absorbed pump energy is fully converted. However, even this overestimated value is still one order of magnitude lower than the density level ($\sim 10^{14} \text{ cm}^{-2}$)[9] required to cause a Mott-transition at 70 K, let alone the level at room-temperature. Furthermore, PAE should also enhance carrier generation in the detuned systems. However, in our experiments, only the tuned system shows a large bandgap renormalisation (Fig.2b and 2d). Hence there must be other mechanisms that can enhance carrier population in addition to PAE. ”

with references:

“[9] Alexey Chernikov, Claudia Ruppert, Heather M. Hill, Albert F. Rigosi, and Tony F. Heinz., Population inversion and giant bandgap renormalization in atomically thin WS₂ layers. Nat. Photon., 9, 466, 2015.”

Furthermore, have added a subsection in Section 6 in the SI including Fig.S13.

In addition, Reviewer #2 has raised some technical issues:

(1) Reviewer #2 suggests that we justify “*fast and repeated hot electron population*” (aka *strong coupling*)” in our system.

Author response: Please see above the answers for point 1.

(2) & (3) Reviewer #2 suggests that we add colour codes in Fig.1b and Fig.3a.

Author response: We are grateful that the reviewer has carefully read our manuscript and we have therefore added colour codes in these figures.

(4) Reviewer #2 suggests that we clarify why data in Fig.3c and 3d look noisy while the intensity plot in Fig.3a looks clean, as they are plotted from the same set of data.

Author response: We have checked these data and plots, finding that no filtering or smoothing have been applied to them. The inconsistency in noise level might be from different axis range taken. Please note that

panel **c** and **d** only show spectra with an energy range from 1.5 to 1.9 eV, while panel **a** shows the intensity plot from 1.6 to 2.5 eV. The denser distribution of data point may reduce the illustration of noise.

(5) Reviewer #2 suggests that we better explain the model (Eq.2) and elucidate our arguement of “coherent hot electron doping” with more thorough analysis.

Author response: We fully agree with the reviewer and therefore have redeveloped our model, having included several major loss factors that may suppress the hot electron doping. By doing so, we have been able to obtain a more accurate estimation of the hot electron density in the WS₂ lattice. As a result, we find that with major losses included, the hot electron density can still reach the level that is required to develop a large bandgap renormalisation as observed in our experiments. Specifically, we have added a new paragraph in the main text (line 173ff):

“To prove that hot electron doping can induce the observed bandgap renormalisation, we need to understand how the hot electrons are generated in our system as well as quantify the net carrier density in lattice. As explained before, due to the off-resonance frequency of pump, plasmons in the PC-WS₂ system can only be effectively excited by coupling to excitons. Specifically the pump energy is absorbed by the semiconductor and down-converted to excite the plasmon-exciton polaritons, which, as half-plasmon half-exciton hybrid states, naturally excite their plasmonic component and result in the generation of plasmonic hot electrons. These charges then overcome the tunneling barrier ($\Delta\phi_{TB}$) formed at the Ag-Al₂O₃-WS₂ interface to dope the WS₂ lattice. During this process, the hot electron doping is subject to several major losses, including (i) the limited pump absorption by the WS₂ MLs, (ii) the losses in energy down-conversion and (iii) the losses due to the hybrid nature of polaritons.”

Following this process, as described in the response to Reviewer #1, we have developed a new model to numerically estimate the density (N_e).

$$N_e = \frac{F_{\text{pump}} \cdot \eta_A \cdot \eta_D \cdot \eta_{\text{pl}}}{2c\epsilon_0} \cdot \mathcal{F} \cdot \frac{1}{\pi^2} \frac{e^2 E_F^2}{\hbar} \frac{\hbar\omega - \Delta\phi_{TB}}{(\hbar\omega)^4} \quad (2)$$

Relevant explanations are given at line 188ff in the main text:

“...Here we take $\eta_A \approx 55\%$ as the absorption coefficient, $\eta_D \approx 66\%$ as the energy down-conversion ratio and $\eta_{\text{pl}} \approx 50\%$ as the excitation ratio of the plasmon component in polaritons. As a result, $F_{\text{pump}} \cdot \eta_A \cdot \eta_D \cdot \eta_{\text{pl}}$ corresponds to the process that the pump energy is absorbed, down-converted and coupled to the plasmonic components in polaritons with major losses included. As optical modes, the excited polaritons gain a spatial distribution (Fig.4b) at the tuned frequency, spreading over the Ag cap surface with hot spots at the interstices between caps. This can be mathmatically expressed as $\mathcal{F} = |\mathbf{E}/\mathbf{E}_0|^2 \dots$ ”

Here we want to thank Reviewer #2 that the efficiency parameters η_A and η_{pl} are all inspired by her/his suggestions, which are the absorption coefficient and the mixing coefficient respectively. Please see above the suggestions for more details. As a result, we have been able to calculate the carrier density, which is stated in the main text (line 200ff):

“Using Eq.2, we are able to plot the spatially distributed hot electron density in the WS₂ monolayer (Fig.4c). The density naturally acquires identical distributions as do the plasmonic excitations, exhibiting inhomogeneous distribution over the area. It has values typically higher than $1 \times 10^{13} \text{ cm}^{-2}$ in most of the

areas, peaking at the interstices between caps with maxima larger than $2 \times 10^{14} \text{ cm}^{-2}$”

In addition, a more detailed discussions of these major losses and the deduction of the revised Eq.(2) have been added into the SI. Specifically, a new sub-section “**Including losses**” has been added in the Section 6 of the SI, which includes Fig.S13 and relevant discussions.

Reviewer #3

We are very happy that Reviewer #3 thinks highly of our manuscript, e.g. “*Ultrafast control of bandgap renormalization can have important applications in ultrafast photonics such as ultrafast optical switch. As such, the results are significant.*” In addition, the reviewer has raised a few issues:

1. Reviewer #3 suggests that we figure out the binding energy of a WS₂ monoalyer deposited on the plasmonic crystal, as this could be largely different from the value for monolayers deposited on a insulating substrate, which will be critical in determining the magnitude of bandgap renormalisation.

Author response: We fully agree with the reviewer and very much appreciate this suggestion. After carefully reviewing literature, we have found that the binding energy of a WS₂ monolayer can be reduced to a half of its original value when deposited on a metal substrate. We have therefore revised our conclusion that the bandgap renormalisation can achieve up to $\sim 550 \text{ meV}$ (about 100 meV smaller than the previous estimation).

Specifically, we have corrected the value in the following sentence in the main text (line 118ff):

“.....It means that in our experiments, the renormalised bandgap starts at $E_g \approx 1.60 \text{ eV}$, lying $\sim 400 \text{ meV}$ below LP and $\sim 550 \text{ meV}$ below the bandgap of WS₂ MLs (given that the binding energy of exciton A is decreased to $\sim 100 \text{ meV}$ when deposited on metal substrates[28], i.e. about a half of the initial value[19]).”

with references:

“[19] Paul D. Cunningham, Aubrey T. Hanbicki, Kathleen M. McCreary, and Berend T. Jonker. Photoinduced Bandgap Renormalization and Exciton Binding Energy Reduction in WS₂, ACS Nano, **11**, 12601, 2017.”

“[28] Park Soohyung, Mutz Niklas, Schultz Thorsten, Blumstengel Sylke, Han Ali, Aljarb Areej, Li Lain Jong, List-Kratochvil Emil J.W., Amsalem Patrick, and Koch Norbert. Direct determination of monolayer MoS₂ and WSe₂ exciton binding energies on insulating and metallic substrates. 2D Mater, **5**, 025003, 2018.”

With the correction, the main conclusion in the manuscript remains unchanged, because $\sim 550 \text{ meV}$ renormalisation is still a large one. But here we want to deeply thank Reviewer #3. Without her/his suggestion, we would have made a serious mistake on the magnitude of bandgap renormalisation.

2. Reviewer #3 suggests that we clarify the method to determine the spectral position of bandgap, since using transient absorption peaks to tell the bandgap position may not apply to 2D semiconductors.

Author response: We agree with the reviewer and have rechecked our measurements for finding the bandgap. Specifically, we have followed the method used in a previous work [Heinz et al., Nat.

Photon., **9**, 466, 2015], using the onset of a spectral feature (i.e. the low energy end) but not the peak to determine the band edge position. Using this method, we can always find the point with the lowest energy, avoiding misjudgement of a bandgap position induced by the not fully filled phase-space. In particular, we have added the following sentence in the main text (line 116ff):

“The onset of the new bandgap can be extracted from the low-energy end of the broad maximum[9] (red dashed vertical line in Fig.4c)”

with a reference:

“[9] Alexey Chernikov, Claudia Ruppert, Heather M. Hill, Albert F. Rigosi, and Tony F. Heinz. Population inversion and giant bandgap renormalization in atomically thin WS₂ layers. Nat. Photon., 9, 466, 2015.”

3. Reviewer #3 suggests that we should discuss the potential implications of hot electron doping, since it is different from the case in Ref[9], where the optical doping (dissociated electron-hole pairs induced by a strong optical pump) is the dominant mechanism.

Author response: We fully agree with the reviewer. We therefore have added new discussions and a reference to discuss the implications of hot electron doping. Specifically, we have added the following sentence in the main text (line 212ff):

“...However, the injected carriers in our system are hot electrons, which are different from the dissociated electron-hole pairs induced by pure optical pumping[9,26], but are more similar to free charge carriers by electrical injection[8,39], where a ~ 550 meV bandgap renormalisation in WS₂ MLs can occur at the electron density of $3 \times 10^{13} - 1.1 \times 10^{14} \text{ cm}^{-2}$ [8]. These results share high similarity with our observation, suggesting that the hot electron doping in our system is able to achieve the threshold to induce a bandgap redshift up to ~ 500 meV with carriers draining from conduction band K to Σ valley[38] that renders the semiconductor indirect”

with references:

“[8] Alexey Chernikov, Arend M. Van Der Zande, Heather M. Hill, Albert F. Rigosi, Ajanth Velauthapillai, James Hone, and Tony F. Heinz. Electrical Tuning of Exciton Binding Energies in Monolayer WS₂. Phys. Rev. Lett., 115, 126802, 2015.”

“[9] Alexey Chernikov, Claudia Ruppert, Heather M. Hill, Albert F. Rigosi, and Tony F. Heinz. Population inversion and giant bandgap renormalization in atomically thin WS₂ layers. Nat. Photon., 9, 466, 2015.”

“[26] D. Erben, A. Steinhoff, C. Gies, G. Schönhoff, T. O. Wehling, and F. Jahnke. Excitation-induced transition to indirect band gaps in atomically thin transition-metal dichalcogenide semiconductors. Phys. Rev. B, 98, 035434, 2018.”

“[38]F. Lohof, A. Steinhoff, M. Florian, M. Lorke, D. Erben, F. Jahnke, and C. Gies. Prospects and Limitations of Transition Metal Dichalcogenide Laser Gain Materials. Nano Lett. 19, 210, 2019.”

“[39] Qiu Zhizhan, Trushin Maxim, Fang Hanyan, Verzhbitskiy Ivan, Gao Shiyuan, Laksono Evan,

Yang Ming, Lyu Pin, Li Jing, Su Jie, Telychko Mykola, Watanabe Kenji, Taniguchi Takashi, Wu Jishan, Neto A H Castro, Yang Li, Eda Goki, Adam Shaffique and Lu Jiong. *Giant gate-tunable bandgap renormalization and excitonic effects in a 2D semiconductor. Sci. Adv.*, **5**, eaaw2347, 2019.”

In addition, Reviewer #3 suggests that we should compare the carrier population in our system with that in the ref[9] in a fairer ground but not only based on a fluence comparison.

Author response: We agree with the reviewer and have therefore directly compare the carrier density. Specifically, we have added a new paragraph in the main text (line 129ff):

“In our system, the carrier density may be increased by plasmonic absorption enhancement (PAE), which, however, can not provide enough carrier population according to our calculation. Specifically, the excitation of plasmons can enhance the absorption of pump by the system, which naturally results in an elevation of carrier numbers in the lattice. In the PC-WS₂ system, the pump intensity can be amplified at the position of the WS₂ ML (Fig.S13 in SI), which, according to our calculation, gives a ~5 times average increase of absorption in the semiconductor. As a result, the carrier density can achieve up to $\sim 1.2 \times 10^{13} \text{ cm}^{-2}$ if the absorbed pump energy is fully converted. However, even this overestimated value is still one order of magnitude lower than the density level ($\sim 10^{14} \text{ cm}^{-2}$)[9] required to cause a Mott-transition at 70 K, let alone the level at room-temperature. Furthermore, PAE should also enhance carrier generation in the detuned systems. However, in our experiments, only the tuned system shows a large bandgap renormalisation (Fig.2b and 2d). Hence there must be other mechanisms that can enhance carrier population in addition to PAE. ”

with references:

“[9] Alexey Chernikov, Claudia Ruppert, Heather M. Hill, Albert F. Rigosi, and Tony F. Heinz., Population inversion and giant bandgap renormalization in atomically thin WS₂ layers. Nat. Photon., **9**, 466, 2015.”

REVIEWER COMMENTS

Reviewer #1 (Remarks to the Author):

All my concerns were addressed properly. In principle, the manuscript can be published now.

Reviewer #2 (Remarks to the Author):

The authors have responded in great detail to the questions and comments of the Reviewers. They have also added much helpful additional analysis.

I first want to specifically address the point of "strong coupling".

1. The authors show, in Fig. S1a angle-resolved transmission spectra of the bare PC sample. This shows two transmission peaks in the wavelength range of the A and B excitons. Out of these two modes, only one of the modes is considered in the coupling scenario that is shown in Fig. S2c. I do – honestly - believe that it is an oversimplification to neglect the existence of these two modes in the coupling scenario. A simple 2x2 model does not work. The same holds for the WS2 sample. Both Xs should be considered in a relativistic coupling scenario, which should be based, at least, on the analysis of a 4 x 4 matrix.

2. If I understand it correctly, the "coupled Lorentz oscillator" model that is used by the authors is, again, oversimplified. I see in Fig. S2b that the spectra are fit two to Lorentz resonances with absorptive lineshape. In the experimental spectra, however, distinct signatures of Fano resonances are seen (for example in the spectra in fig. S1 at 40° around the XA exciton). Such Fano resonances arise if the emission of a broader resonance interferes with the emission from a narrower resonance. Such Fano-type lineshapes are neglected in the analysis (if I understand it correctly), even though they are evidently seen in experiment. I think that it is necessary to really calculate the linear transmission spectra for the discussed 4x4 model if the authors want to gain a detailed understanding of the optical properties of their sample. For these reasons, I am skeptical about the "blue bullets" shown in Fig. 2c (and the normal mode splitting deduced from them). I have the suspicion that the actual normal mode splitting may turn out to be smaller when analyzing a more appropriate 4x4 model.

3. In their analysis, the authors conclude (page 4 of the SI, 2nd par) that $\kappa > 2g > \gamma$. This implies that the sample is *not* in the strong coupling regime but in an intermediate coupling regime. I would therefore very much suggest to omit the claim that the sample is in the strong coupling regime. This holds in particular since the 2x2 model that is used for extracting the parameters κ , g and γ is oversimplified. I therefore firmly suggest to remove the claims that the sample is in the strong coupling regime. Otherwise, a complete lineshape analysis, including all relevant resonances in the sample, together with a demonstrating that strong coupling is truly reached, is necessary. I also point out that I believe that the conclusions drawn from the pump-probe measurements (which cannot resolve the Rabi oscillations anyway) will not change if the authors admit that the sample is "only" in the intermediate coupling regime.

4. The authors probe a normal mode coupling of many excitons to several plasmon resonances. This "classical harmonic oscillator coupling" is different from a Vacuum Rabi splitting in the optical spectra of one single quantum emitter coupled to a single cavity mode. The term "Vacuum Rabi splitting" should therefore be avoided.

In summary, I think that the manuscript would greatly benefit from a much more thorough analysis of the linear optical transmission spectra in Fig. S1. Such an analysis must include all relevant spectral resonances (in the present case it seems that this requires at least a 4x4 model). I still believe that the results may be of interest for the audience of Nature Communications.

Reviewer #3 (Remarks to the Author):

The authors have adequately addressed my comments and improved their manuscript. I recommend acceptance of the revised manuscript.

Reviewer #2

We are glad that Reviewer #2 noted our efforts in improving the manuscript and we are happy that s/he “*still believe the results may be of interest for the readers of Nature Communications*”. In responding to her/his new suggestions, we figure that the problems in our system have now been much better addressed and the role of plasmon-exciton coupling has been further clarified. Thank you!

In general, Reviewer #2 suggests that we should perform more detailed analyses on the transmission spectra and the coupling state of our PC-WS₂ system. Specifically,

1. Reviewer #2 first suggests that we should analyse the transmission spectra using a coupled model that includes at least 4 oscillators, because there are 4 sets of resonances co-existing and mutually interacting in our system, which are PC-01 mode, PC-02 mode, exciton A and exciton B, respectively.

2. In addition to point 1, Reviewer #2 suggests that we should study the optical spectra using a more sophisticated model than the current one, which may better explain the special lineshapes, e.g. Fano-like lineshapes at oblique incident angles, so that the resonance dispersions can be more accurately reproduced.

Author response: Since point 1 and 2 are very relevant, here we respond them together. First of all, we are grateful for these suggestions, which enable us to gain a more comprehensive understanding of our own system. We fully agree with the reviewer on these points and have made relevant revisions.

(1) We have used the transmission coefficients that build on 4 coupled Lorentz oscillators to reproduce the spectral lineshapes, which is:

$$T(\omega) = |t(\omega)|^2 = \left| a + \sum_{j=1}^4 b_j \frac{\pi \frac{\gamma_j}{2}}{(\omega - \omega_j) - i \frac{\gamma_j}{2}} \right|^2 \quad (\text{S1})$$

As compared to our previous model, i.e. the linear superposition of the amplitudes of component Lorentz oscillators, Eq.S1 allows us to calculate the transmission spectra that result from the interplay between all 4 oscillators in our system. As the result, we are able to well reproduce the spectral lineshapes from all incident angles, including the Fano-like lineshape at 40°. Fig.S2(b) exhibits examples of these fittings. Then the spectral positions of the fitted resonances were extracted and plotted as a function of angles, shown as the blue dots in Fig.S2(c).

(2) We then used a 4 × 4 matrix to fit the dispersion of these blue dots, which is:

$$\begin{pmatrix} \tilde{E}_{\kappa_1}(\theta) & g_{1A} & 0 & g_{1B} \\ g_{1A} & \tilde{E}_{\gamma_A} & g_{2A} & 0 \\ 0 & g_{2A} & \tilde{E}_{\kappa_2}(\theta) & g_{2B} \\ g_{1B} & 0 & g_{2B} & \tilde{E}_{\gamma_B} \end{pmatrix} \begin{pmatrix} \alpha_{\kappa_1}(\theta) \\ \alpha_{\gamma_A}(\theta) \\ \alpha_{\kappa_2}(\theta) \\ \alpha_{\gamma_B}(\theta) \end{pmatrix} = \tilde{E}_p(\theta) \begin{pmatrix} \alpha_{\kappa_1}(\theta) \\ \alpha_{\gamma_A}(\theta) \\ \alpha_{\kappa_2}(\theta) \\ \alpha_{\gamma_B}(\theta) \end{pmatrix} \quad (\text{S2})$$

Using Eq.S2, we have been able to plot the curves [orange curves in Fig.S2(c)] that follow the dispersions of the fitted resonances. We note that the fitted dispersion curves agree well with spectral positions near the intersection point between PC-01 mode and exciton A, exhibiting a split spectral feature. In contrast, the curves slightly drift from the fitted resonances near the intersection point between PC-02 mode and exciton B. We think it is due to the lossy nature of the coupled oscillators, e.g. both PC-02 mode and exciton B

Figure.S2 (SI): Analysis of spectral line shape and plasmon-exciton coupling.

acquire broad linewidths, which may temper the fitting accuracy using Eq.S1.

(3) From the fitting, we've learned that the strength of coupling between PC-01 and exciton A is $g_{1A} \approx 87meV$ and the strength of coupling between PC-02 and exciton B is $g_{2B} \approx 30meV$. What's worth noting is that there is also a coupling between PC-01 mode and exciton B. This can be seen from the blue shift of the resonances near exciton B at low incident angles ($\theta = 0 - 6^\circ$), yielding a strength of $g_{1B} \approx 70 meV$. Here we want to point out that we wouldn't have been able to find this 01-B coupling without using the 4×4 coupled oscillator model. **Therefore we very much appreciate this suggestion from Reviewer #2.**

Detailed revisions corresponding to these two points can be found in Section 1 of supplementary information (SI).

3. Reviewer #2 suggests that we should define our system as in "intermediate" coupling state, if the coupling strength only meets $2g > (\kappa + \gamma)/2$ but does not satisfy $2g > \kappa > \gamma$.

Author response: We fully agree with this point, as an accurate identification of coupling state will provide clear application conditions for the coherent doping of plasmonic hot electrons, which is critical for this work. As mentioned above, the coupling strength $g_{1A} \approx 87meV$, which satisfies $2g_{1A} > (\kappa_1 + \gamma_A)/2$, yet

is slower than the plasmonic dephasing. Therefore, the coupling between PC-01 mode and exciton A should be defined as intermediate coupling. We have added this new definition and removed the previous “strong coupling” definition throughout the manuscript including the main text and SI.

4. Reviewer #2 suggests that we should avoid the term “Vacuum Rabi splitting”, as this term is specially used to describe the coupling between a single emitter and a single cavity photon, which is different from our case, in which multiple plasmons couple with multiple excitons.

Author response: We fully agree with Reviewer #2 on this point, and therefore have replaced “Vacuum Rabi splitting” as “spectral splitting”.

In addition to the specific points raised by the reviewers we have also revised some other parts in the manuscript, e.g. abstract, some paragraphs and figure captions.

REVIEWERS' COMMENTS

Reviewer #2 (Remarks to the Author):

Dear authors, I quickly looked at your response. I am very happy to read that you have now performed such a thorough analysis of your data. I believe that this new analysis gives a physically much more accurate description of your data.

I also acknowledge the removal of claims on "strong coupling" and VRS. This certainly improves the quality of the paper.

If I may, I would like to suggest to add Fig. S2 to the main manuscript. It is crucial for understanding the time-resolved results and will certainly be of interest for many readers.

I gladly recommend your manuscript for publication in Nature Comm.
Christoph Lienau

Reviewer #2

We are very much glad that Reviewer #2 recommended our manuscript to be published in *Nature Communications*. Thank you!

In addition, Reviewer#2 suggests that we add Supplementary Figure 2 into the main text, which will help readers better understand our system. We fully agree on this point and have integrated the main contents, i.e. panel (b) and (c), of Supplementary Figure 2 into Figure 1 in the main text. Please see below.

Figure.1 (Main Text): Sample structure and analysis of spectral line shape and plasmon-exciton coupling.

In addition, we have adapted the main text accordingly to explain the newly added panel (c) and (d) of Figure 1. Specifically, we have added the following text (line 69ff):

“As a result, within the angle range (0–40°), PC-01, PC-02, X_A and X_B mutually interact. To analyse the couplings between them, we have used a coupled Lorentz model that build on 4 sets of Lorentz oscillators (Supplementary Eq. S1) to fit the transmission spectra of the PC-WS₂ sample. Fig. 1c shows examples of these fittings at different incident angles. The spectral positions of the fitted resonances were then extracted and plotted as a function of angles (blue dots in Fig. 1d). The complicated dispersive behaviours of these resonances were then fitted using a (4 × 4) matrix of coupled oscillators (Supplementary Eq. S2) to give the critical coupling parameters.”

In addition to the specific points raised by the reviewers we have also revised some other parts in the manuscript, e.g. abstract, some paragraphs and figure captions.